# Sb_2_S_3_-Based Dynamically Tuned Color Filter Array via Genetic Algorithm

**DOI:** 10.3390/nano13091452

**Published:** 2023-04-24

**Authors:** Xueling Wei, Jie Nong, Yiyi Zhang, Hansi Ma, Rixing Huang, Zhenkun Yuan, Zhenfu Zhang, Zhenrong Zhang, Junbo Yang

**Affiliations:** 1Guangxi Key Laboratory of Multimedia Communications and Network Technology, School of Computer, Electronic and Information, Guangxi University, Nanning 530004, China; 2113391114@st.gxu.edu.cn (X.W.); 2113391106@st.gxu.edu.cn (J.N.); 2113391081@st.gxu.edu.cn (Y.Z.); 2113391094@st.gxu.edu.cn (R.H.); 2013391132@st.gxu.edu.cn (Z.Y.); 2Center of Material Science, National University of Defense Technology, Changsha 410073, China; 201821521328@smail.xtu.edu.cn (H.M.); zhenfuzhang@nudt.edu.cn (Z.Z.)

**Keywords:** structural color, genetic algorithm, inverse design, phase change material, Mie resonance

## Abstract

Color displays have become increasingly attractive, with dielectric optical nanoantennas demonstrating especially promising applications due to the high refractive index of the material, enabling devices to support geometry-dependent Mie resonance in the visible band. Although many structural color designs based on dielectric nanoantennas employ the method of artificial positive adjustment, the design cycle is too lengthy and the approach is non-intelligent. The commonly used phase change material Ge_2_Sb_2_Te_5_ (GST) is characterized by high absorption and a small contrast to the real part of the refractive index in the visible light band, thereby restricting its application in this range. The Sb_2_S_3_ phase change material is endowed with a wide band gap of 1.7 to 2 eV, demonstrating two orders of magnitude lower propagation loss compared to GST, when integrated onto a silicon waveguide, and exhibiting a maximum refractive index contrast close to 1 at 614 nm. Thus, Sb_2_S_3_ is a more suitable phase change material than GST for tuning visible light. In this paper, genetic algorithms and finite-difference time-domain (FDTD) solutions are combined and introduced as Sb_2_S_3_ phase change material to design nanoantennas. Structural color is generated in the reflection mode through the Mie resonance inside the structure, and the properties of Sb_2_S_3_ in different phase states are utilized to achieve tunability. Compared to traditional methods, genetic algorithms are superior-optimization algorithms that require low computational effort and a high population performance. Furthermore, Sb_2_S_3_ material can be laser-induced to switch the transitions of the crystallized and amorphous states, achieving reversible color. The large chromatic aberration ∆E modulation of 64.8, 28.1, and 44.1 was, respectively, achieved by the Sb_2_S_3_ phase transition in this paper. Moreover, based on the sensitivity of the structure to the incident angle, it can also be used in fields such as angle-sensitive detectors.

## 1. Introduction

Structural color refers to the color generated by modulating the light propagation through the reflection, scattering, and interference of light between micro-nano structures and specific wavelengths [1,2,3,4]. Structural color provides significant benefits of high durability, environmental friendliness, and high resolution, compared with toxic and volatile traditional chemical dyes [5,6,7]. In addition, structural color is compatible with complementary metal-oxide-semiconductor (CMOS) fabrication processes [8]. The inherent loss of plasma in the metal surface plasmon resonance (SPR) significantly limits the saturation and brightness of the resulting structural colors [9]. The Fabry–Perot color filters [10,11], due to their potential for enhancing color purity and gamut, have been the subject of extensive investigations by numerous scholars. Additionally, the use of dielectric nanostructures [12] that rely on Mie resonance offers a promising approach for developing optical systems that are both lightweight and low-loss. Dielectric nanostructures function as resonators at optical frequencies by supporting electric and magnetic dipoles as well as higher-order modes [13,14] and reflect visible light at different wavelength bands via Mie resonance. As such, they have successful applications in holograms [15,16], anti-counterfeiting [17,18,19], metalenses [20], and sensors [21,22].

The design of most structural color devices is “static”, thereby limiting their practical application. In contrast, tunable color devices are more desirable for display applications [23]. Generally, the color of tunable dielectric nanostructures [24,25] can be classified into inorganic and organic types. Compared with organic types, inorganic tunable colors have the advantages of a high cycle rate, high stability, and more durability. In particular, phase change materials (PCMs) have garnered significant attention in the field of inorganic tunable color due to their reversible phase transition, low switching energy, and nanosecond-level switching speed [26,27,28,29,30]. Commonly used PCMs include VO_2_, Ge_2_Sb_2_Se_4_Te (GSST), and Ge_2_Sb_2_Te_5_ (GST), and each of these has distinct optical properties, as summarized in Table 1. VO_2_ exhibits a low transition temperature of 68 °C and a high switching speed, but its metallic state is unstable and volatile at room temperature [31,32]. Additionally, it requires constant energy to maintain its optical properties, making it challenging for optical storage applications. GSST [33] has 10^5^ phase transition cycles but has a high conversion temperature of 400 °C. Additionally, it has a relatively small contrast of the real part of the refractive index at visible light frequencies (∆Re(n) = 0.16) but has a high extinction coefficient of 3.82, which makes it unsuitable for use in the visible wavelength band. Among various PCMs, GST is the most commonly used phase change material in integrated photonics, as it can perform nanosecond transitions between the crystallized and amorphous states and repeat 10^15^ cycles [34], and phase transitions are inherently non-volatile. However, the performance achieved by using GST to tune optical devices in the visible light band is not ideal [35] due to its band gap ranging between 0.5 and 0.7 eV in the visible light band, high absorption, and relatively small contrast of ∆Re(n). The use of an ultra-thin GST layer can reduce excessive absorption in the visible light band but further limits the optical path length change caused by the phase transition, resulting in limited device size and scalability.

This paper introduces Sb_2_S_3_ as a potential PCM for optical applications. With a wide bandgap of 1.7–2 eV in the visible light band, Sb_2_S_3_ exhibits lower propagation loss than GST when integrated into a silicon waveguide. Figure 1 presents the refractive indices of Sb_2_S_3_ and GST. Sb_2_S_3_ exhibits nonvolatile optical properties, requires energy only during the switching process, and possesses a switching temperature and speed comparable to GST, both being in the nanosecond time scale [36]. In terms of integrated device configurations, Sb_2_S_3_ exhibits a closer proximity to the real part of the refractive index of its crystallized and amorphous phases to silicon, thus facilitating mode matching. Furthermore, the deposition process of Sb_2_S_3_ is compatible with CMOS technology. Sb_2_S_3_ can be prepared by various methods, such as spray pyrolysis [37], electrodeposition [38], chemical deposition [39,40], and thermal vacuum evaporation [41,42]. Phase transformation can be switched precisely via semiconductor laser heating [43] and the electric pulse method of the heating device [32]. The cycle durability of Sb_2_S_3_ can be enhanced by reducing the energy of the pulse and increasing the number of pulses. Kun Gao et al. [44] achieved 7000 cycles through the combined control of pulse energy and pulse number. Li Lu et al. [15] demonstrated that Sb_2_S_3_ can effectively tune the Mie resonance in visible light and proposed an experimental method for the reversible switching of thick Sb_2_S_3_ nanodisks. Overall, Sb_2_S_3_ has been deemed a more suitable phase change material for tuning visible light than GST.

The majority of studies on dielectric nanostructure rely on the traditional forward-design approach, which can only optimize a single parameter and is limited in its ability to tune parameters, solve complex problems, and display color. However, the Genetic Algorithm (GA) can overcome these limitations by mimicking natural selection to find the optimal parameter set for complex non-analytic problems. Therefore, it offers certain advantages in designing complex structures with specific functions. In this study, we demonstrate the inverse design of nanostructures using the genetic algorithm [45,46,47,48], which allows for the input of desired target colors. This approach significantly reduces the time and cost associated with the design process and improves the performance of the metasurface structures compared to the traditional forward-design approach.

In this paper, we present the inverse design of Sb_2_S_3_-based nanoantennas for tunable structural color using a combination of the genetic algorithm and finite-difference time-domain (FDTD) solutions. The Sb_2_S_3_-based nanoantennas are designed as rings or cylinders and deposited on the multilayer film structure composed of the Ag layer and the SiO_2_ layer. The proposed method achieves the tunable properties of dielectric nanostructures through the phase transition of Sb_2_S_3_ material. By fabricating dielectric nanoantennas directly composed of Sb_2_S_3_, high-efficiency devices with nonvolatile characteristics can be realized. Through the intelligent search of the algorithm, we found a set of structural parameters that are most suitable for the target color and realized a large chromatic aberration ∆E modulation of 64.8, 28.1, and 44.1, which theoretically proves the effectiveness and feasibility of the design.

## 2. Methods

### 2.1. Metasurface Structure

The metasurface structure is composed of many identical unit structures, as displayed in Figure 2a. The schematic of the unit structure’s cross-section is demonstrated in Figure 2b. The dielectric nanoantenna made of Sb_2_S_3_ is represented by the beige, which is deposited on the multilayer film structure composed of the Ag layer and the SiO_2_ layer. The dielectric nanoantennas, fabricated from Sb_2_S_3_, may come in the form of rings or cylinders that are controlled by the algorithm to enable a high degree of flexibility in parameter design, thereby enhancing the metasurface’s designability. The bottom of the metasurface structure is an Ag layer with a fixed thickness of 100 nm, which is utilized to block visible light transmission and improve the structure’s reflection capacity. The SiO_2_ layer’s thickness is also fixed at 200 nm to preserve the structure’s resonance characteristics.

### 2.2. Process of Inverse Design

Micro-nano structures are capable of generating colors by supporting electric dipoles, magnetic dipoles, and higher-order modes. The geometry of metasurfaces is closely related to dipole resonance, making it necessary to optimize the geometry. In traditional design processes, a regular initial structure is obtained through prior methods, and subsequently, the structure is designed by manually selecting parameters with the goal of achieving desirable spectral effects. As illustrated in Figure 3, in the traditional design process, the known variables of the structure are manually selected; thus, structures with different parameters are obtained. The spectral properties corresponding to each structure are then obtained through a forward design process, which involves obtaining the structural dependent variable through the structural independent variable. However, this design process leads to the limited parameter space that can be adjusted by traditional design methods, which can be time-consuming and requires significant effort, particularly for complex structures with large parameter spaces and high degrees of freedom. In contrast, inverse design methods are commonly employed to obtain the corresponding structure from the desired spectral properties. For example, S. S. Panda et al. [45] utilized the genetic algorithm to inverse-design an eight-fold symmetric polygonal unit structure for the implementation of an all-dielectric metasurface filter array and X. Huang et al. [17] employed the genetic algorithm to inverse-design irregular polygonal cell structures. Such works are characterized by structural complexity, and forward design approaches may be inadequate to solve such problems. While algorithm optimization can not only improve the performance of the metasurface spectral filtering structure but also endow the metasurface with a higher degree of freedom and a larger search space. Forward and inverse design methodologies each possess distinct merits. Relatively speaking, inverse design has certain advantages for designing complex structures to realize specific functional devices.

The genetic algorithm is an optimization procedure based on the simulation of natural heredity and the selection of survival of the fittest. It deals with complex problems through selection, crossover, and variation using internal chromosome (gene) information and gradually approaches the optimal solution through the genetic operator, which is a superior-optimization algorithm that is adept at finding the optimal solution. Compared with the gradient optimization algorithm, the genetic algorithm is a search algorithm with parallelism, randomness, and directionality [7]. Meanwhile, it exhibits probabilistic numerical search and data attributes that offer significant advantages in multi-objective optimization. In this study, we utilize the genetic algorithm to optimize the inner diameter *r*, outer diameter *R*, height *H* of the ring, and substrate’s period *P*. These four parameters are used as the input variables of the algorithm, and each group of input variables represents a gene individual. We set the target wavelength and objective function and it evolves through repeated operations of the gene interaction mechanism to find the relationship between the structure geometry and the reflection spectrum. The optimization process continues until the fitness value of the optimal individual in the population no longer improves. The specific optimization process of inverse design using the genetic algorithm is shown in Figure 4.

In this paper, the quality of the individuals is evaluated by the FOM function, which is used as a measure of the fitness of genetic individuals to find the optimal solution:(1)$$FOM=FOM_{ad}/1000+FOM_{cf}$$
(2)$$FOM_{ad}=|i-peak(R_{a})|$$
(3)$$FOM_{cf}=\varepsilon R(i)-\sum_{1}^{i-1}R(\gamma )/(i-1)-\sum_{i+1}^{301}R(\gamma )/(301-i)$$
where $FOM_{ad}$ represents the distance between the reflection peak position and the target wavelength in the amorphous state, and the larger the value of $FOM_{ad}$, the more significant ∆E can be generated. To generate color with high purity, it is necessary to suppress resonance outside the target wavelength. $FOM_{cf}$ indicates the saturation of the color represented by the crystallized state at the target wavelength and the evaluation of the degree of secondary wave suppression. i is the target wavelength, $peak(R_{a})$ indicates the position of the peak structural reflection in the amorphous state, R(i) denotes the reflectance at the target wavelength in the crystallized state, $\sum_{1}^{i-1}R(\gamma )$ and $\sum_{i+1}^{301}R(\gamma )$ represent the sum of reflectance at wavelengths other than the target wavelength, and these two parts are averaged, respectively. The imbalance weight ε is set to maximize the reflection peak at the target wavelength and to suppress the resonance peaks in other wavelengths that affect the color purity.

### 2.3. Color Calculation

The color properties of the metasurface structures are evaluated by converting the reflectance spectra obtained from FDTD simulations into color representations. Any color can be represented by tristimulus values, and the representation of these three components is as follows:(4)X=∫R(η)×CIEX(η)×d(η)
(5)Y=∫R(η)×CIEY(η)×d(η)
(6)Z=∫R(η)×CIEZ(η)×d(η)
where R(η) is the spectrum data monitored from FDTD, CIEX(η), CIEY(η), and CIEZ(η) denote the CIE color matching function (CMF), d(η) is the incident light intensity, and the wavelength of the visible range is selected as 400–700 nm, in this paper. The chromaticity coordinates *x*, *y*, and *z* are obtained by normalizing the *XYZ* tri-stimulus values to a scale of 0–1: (7)x=X/(X+Y+Z)
(8)y=Y/(X+Y+Z)
(9)z=1−x−y

The obtained (*x*, *y*) coordinates can be represented on the CIE 1931 chromaticity diagram. The ∆E in this paper can be quantified via CIE DE2000 as follows: (10)$$L^{*}=116f(\frac{Y}{Y_{n}})-16$$
(11)$$a^{*}=500(f(\frac{X}{X_{n}})-f(\frac{Y}{Y_{n}}))$$
(12)$$b^{*}=200(f(\frac{Y}{Y_{n}})-f(\frac{Z}{Z_{n}}))$$
where
(13)$$\begin{array}{c} f(t)=\left\{\begin{array}{ll} \sqrt[{3}]{t} & t<l^{3}\\ \frac{t}{3l^{2}}+\frac{4}{29} & \;t>l^{3} \end{array}\right.\\ l=\frac{6}{29} \end{array}$$

The ∆E can be determined from *L***a***b** [49].

## 3. Results and Discussion

Figure 5 displays the optimized structure and Table 2 lists the specific parameters for each structure. Upon the incident of light on the filter array, different colors are generated in reflective mode through the Mie resonance inside the metasurface structure, and the reversible tunable characteristics are realized through the phase transition of Sb_2_S_3_ nanoantennas. The reflection spectra of each structure are obtained by the FDTD solutions and are shown in Figure 6. It is worth noting that the high reflection peak at 660 nm and the clutter that affects the pure color is fully suppressed so that the pure color is reflected in the crystallized state. Moreover, although blue light and green light also have higher reflection peaks at the target wavelengths of 450 nm and 545 nm, certain resonances in other wavelength bands affect their pure color reflections. To investigate how the structure influences the optical response of reflected blue and green light colors, the resonance modes of the proposed nanoantenna were evaluated based on the electromagnetic field profile. Figure 6e shows the electromagnetic field profiles of several resonance peaks in the blue and green light reflection spectra. The electromagnetic field profiles show that different structures generated various Mie resonances at different wavelengths, resulting from multi-mode electric and magnetic field coupling. The Mie resonance of blue light at 450 nm is mostly related to the gap between the SiO_2_ layer and the Sb_2_S_3_ nanoantenna, particularly in the SiO_2_ layer. The Mie resonance at 555 nm is associated with the SiO_2_ layer, creating a strong resonance inside the SiO_2_. The resonant peak at 700 nm arises from the Mie resonances at the left and right tops of the Sb_2_S_3_ nanoantenna. The Mie resonances of the two resonant peaks in green light both occur at the Sb_2_S_3_ nanoantennas, and the energy is highly confined between the Sb_2_S_3_ nanoantennas. As the absorption of Sb_2_S_3_ becomes significant in the short-wave band, the reflection efficiency decreases, and other resonances outside the target wavelength are generated, affecting the structure’s reflection performance.

Another optimizing objective of this study is the distance between reflection peaks in different phase states. When the Sb_2_S_3_ nanoantenna changes from the crystallized state to the amorphous state, it is accompanied by a sharp change in the reflectivity curve, and the peak shift amplitude reaches 250 nm, 155 nm, and 40 nm, respectively. The color calculation of the metasurface structure shows that ∆E is 64.8, 28.1, and 44.1, respectively, which provides favorable conditions for strong color contrast, as demonstrated in Figure 6d.

PCMs are currently widely used in the design of dynamic metasurfaces. For example, the GeTe material was proposed by S. G. C. Carrillo et al. [50] for a dynamic display resonant absorber, and X. Huang et al. [17] proposed a guided film resonant reflective device utilizing a thin layer of GST material and a multilayer film structure. A comparison between these two works and the present study is presented in Figure 7. In the crystallized state, the color gamut obtained in this work is significantly wider than the 6.2% color gamut obtained by S. G. C. Carrillo et al., and the achieved ∆E is much larger than the 24.8, 24.9, and 4 obtained in their work. The color gamut obtained in this work is comparable to that obtained by X. Huang et al., but the ∆E is larger than the values of 23.39, 13.77, and 35.42 obtained by them. Furthermore, the irregular design of the structure proposed by X. Huang et al. makes it challenging to use them in daily life. Based on the simulation results and the specific comparison of the performances of devices presented in Table 3, the proposed device’s structure in this study offers superior color gamut and ∆E performance while meeting the requirements of daily use.

In the pursuit of inverse design for structural color, various optimization algorithms have been employed, such as deep learning, gradient descent, and topological optimization, as illustrated in Table 3. In contrast to the gradient algorithm, deep learning and genetic algorithms exhibit probabilistic numerical search and data attributes that offer significant advantages in multi-objective optimization [51]. With the approach suggested by Z. Huang et al. [46], a structural color inverse design technique utilizing machine learning methodologies has been introduced. Specifically, supervised learning models are employed herein to train the geometry and color of dielectric arrays, which are subsequently implemented within a reinforcement learning algorithm to find out the optimal optical structure required for the desired color. This method presents a solution to the issue of non-uniqueness commonly encountered in the inverse design process. In a similar vein, D. Ma et al. [52] presented an approach for multicolor meta-holography utilizing deep learning. Specifically, their work leveraged a hybrid framework incorporating neural networks and evolutionary strategies to achieve a structural inverse design that meets the desired resonant wavelength, bandwidth, and phase delay requirements. Due to the stochastic and hereditary properties inherent in the genetic algorithm, the algorithm in this work also provides a feasible method for realizing the structural color inverse design of the desired resonance wavelength.

**Table 3 nanomaterials-13-01452-t003:** Performance of design devices.

Refs.	[17]	[46]	[50]	[52]	This Work
Material	GST/SiO_2_/Ag	Si/SiO_2_	ITO/GeTe/Al	TiO_2_/dimers/SiO_2_	Sb_2_S_3_/SiO_2_/Ag
Structure	Irregular Grating + Multilayer film	Nanoantenna + SiO_2_ film	Nanoantenna + Multilayer film	Two kinds of anisotropic dielectric nanostructures + SiO_2_ film	Nanoantenna + Multilayer film
Working Bandwidth (nm)	400–700 nm	380–780 nm	400–700 nm	400–800 nm	400–700 nm
Design Method	Genetic algorithm inverse design optimization	Machine learning inverse design optimization	Forward design optimization	Neural network and evolutionary strategy inverse design optimization	Genetic algorithm inverse design optimization
Physical Mechanism	Mie resonance	Mie resonance	MIM resonance	—	Mie resonance
Peak Shift Amplitude (R/G/B)	270/180/230 nm	—	13/18/10 nm (measure)	—	250/155/40 nm
Chromatic Aberration	35.42/13.77/23.39	—	24.8/24.9/4	—	44.1/28.1/64.8
crystallized color gamut (sRGB color space)	36.4%	—	6.2%	—	27.8%

With the optimization of the genetic algorithm, colors with better purity and significant ∆E are obtained. To assess the algorithm’s effect on the color performance of a given structure during the optimization process, we compare the performance of the initial and optimal populations through the reflection spectrum and the color representation in the CIE 1931 color space, as depicted in Figure 8. In Figure 8a–c, it is evident that the phase state variations in the structures elicit distinct resonance responses. Despite the optimal structure obtained through the algorithm not achieving the highest reflection peak at the target wavelength, the algorithm successfully suppresses resonances that interfere with the pure color generation, ultimately generating a more saturated final color, as shown in Figure 8d. Moreover, the convergence process of the FOM function is shown in Figure 8e. Initially, a rapid change can be observed before it tends stabilize toward the end, indicating that the structure’s performance develops positively with the help of the algorithm.

We investigated the trends of the reflectance spectra and color properties of the structures with variations in incident and polarization angles, changing the incidence angle in steps of 5° from 0° to 20°, as shown in Figure 9. The results reveal that, as the incidence angle increased, the reflection peak at 450 nm of blue light in the crystallized state increased from 37% to 48%, and the resonance position gradually blue-shifted from 450 nm to 434 nm. Additionally, the resonance peak at 555 nm red-shifted to 661 nm, while the reflection peak increased from 28% to 55%. We also observe that the change in incident angle resulted in the transition of blue light in the crystallized state to purple light, as shown in Figure 9g. It can be seen from Figure 9b that the resonance peak at 545 nm of green light in the crystallized state gradually blue-shifted with the increase in the incident angle. This led to a gradual decrease in the reflection peak, which eventually coincided with the new resonance peak at 430 nm. Additionally, the resonance at 610 nm gradually red-shifted with the angle increasing, and the peak increased from 16% to 72%. From Figure 9h, we observed that the change in incidence angle led to the transition of green light to blue light in the crystallized state. The results also revealed that the change in the angle of incidence led to a red shift of the resonance peak of the red light and resulted in elevated reflections from other bands, resulting in the color coordinates gradually shifting towards the middle of the color space, as shown in Figure 9c,i. We calculated the effect of the incidence angle on structural ∆E using the above equation, as shown in Figure 10b. As the incidence angle increased from 0° to 20°, the ∆E changed from 64.8, 28.1, and 44.1 to 52.3, 46.9, and 27.9, respectively. The ∆E remained significant and was still much higher than the minimum standard (1.5–3) distinguishable by the human eye. From Figure 9g–i and Figure 10a, it is clear that the reflection of the structure is sensitive to the incident angle and that it has a specific angle dependence, where the incidence angle variation has a more significant effect on the crystallized state than the amorphous state. The angle-dependent characteristics of the device can be applied to angle-sensitive detectors. 

To investigate the sensitivity of the structures to polarization, the scan parameter function within the FDTD solutions was used to obtain the reflection spectra of the three structures with different polarization angles in two-phase states, varying the polarization angle from 0° to 90° in steps of 1°, as shown in Figure 11. Figure 11 indicates that the symmetric shape of the rings or cylinders of the dielectric antennas endows the designed structure with insensitivity to the incident polarization angle.

Finally, we discuss the feasibility of the practical production of the devices and several factors that may affect device performance. The substrates used in the fabrication process consist of multilayer structures with fixed thicknesses, which can be fabricated by utilizing e-beam lithography (EBL) and sputtering deposition techniques [53]. Furthermore, the proposed phase change material, Sb_2_S_3_, can be prepared using various techniques, such as spray pyrolysis [37], electrodeposition [38], chemical deposition [39,40], and thermal vacuum evaporation [41,42]. The phase change process can be switched via semiconductor laser heating [43] or the electric pulse method of the heating device [32]. It has been experimentally demonstrated that the amorphization of thick Sb_2_S_3_ nanodiscs can be effectively performed at 12mW, 780 nm, 100 fs laser pulses, and a scanning speed of 10, as proposed by Li Lu et al. [15]. Additionally, the cycle durability of Sb_2_S_3_ can be improved by controlling the pulse energy and pulse numbers. For instance, Kun Gao et al. [44] achieved 7000 cycles through the combined control of pulse energy and pulse number, indicating its practical potential for mass production. Simulation data of the metasurface are discussed in this paper, and related experimental work is in progress and will be reflected in the subsequent research results.

The deposition of the Si_3_N_4_ encapsulation layer on the device surface prevents sulfur loss in Sb_2_S_3_ materials during annealing and laser switching. [15] Given that the simulations reported in the study did not account for the presence of a 30 nm thick Si_3_N_4_ encapsulation layer, we discuss the effect of adding a Si_3_N_4_ encapsulation layer on the performance of the structure at 660 nm in reflecting red light. Figure 12a depicts the diagram of the unit structure after the addition of a Si_3_N_4_ encapsulation layer, while Figure 12b illustrates the cross-sectional view of the unit posts incorporating a Si_3_N_4_ encapsulation layer with a thickness of 30 nm. The reflection spectra curves of Sb_2_S_3_ in crystallized and amorphous states are obtained, respectively, after adding the upper covering layer, as illustrated in Figure 12c,d. It can be found that the Sb_2_S_3_ exhibits a reduction in reflection peak intensity from 71% to 58%, accompanied by a red-shift in the resonance position from 660 nm to 668 nm in its crystallized state. In contrast, Sb_2_S_3_ demonstrates a blue-shift in the reflection peak in its amorphous state. To more clearly represent the effect of the overlay on device performance, the reflectance spectra are characterized in color on a CIE 1931 chromaticity diagram. The results reveal that, despite the decrease in reflection peak, there is a discernible displacement of the coordinates towards the boundary, thereby enhancing the color saturation, as illustrated in Figure 12e. Using the above equation, calculations confirmed that the ∆E changed from 44.1 to 29.7 after adding 30 nm of Si_3_N_4_ encapsulation layer, where the ∆E is still significant (the minimum standard for the human eye to distinguish ∆E is 1.5 to 3).

L. Lu et al. [15] utilized radio frequency (RF) sputtering to deposit Sb_2_S_3_, followed by electron beam lithography (EBL) for nanopatterning. AA El-Shazly et al. [42] employed the thermal evaporation technique to synthesize Sb_2_S_3_ and effectively managed to modulate its thickness and deposition rates with the aid of a quartz crystallized thickness monitor. RS Mane et al. [40] fabricated Sb_2_S_3_ through a chemical deposition method at a controlled temperature of 6(±2) °C. The Sb_2_S_3_ thickness range of 77–206 nm was obtained via deposition at different times. Their approach demonstrated the tunability of the thickness of Sb_2_S_3_, which may have potential implications in the development of electronic and optoelectronic devices. IS Virt et al. [54] reported the preparation of variable thickness in Sb_2_S_3_ using the pulsed laser ablation technique. The deposition process allows for the precise control of thickness to obtain Sb_2_S_3_ with a thickness ranging from 40 nm to 1500 nm. Given that the gratings employed in this study possess varying thicknesses, an analysis of the impact of deviations in the fabricated thickness on the device’s overall performance is presented in this paper. We performed a scanning analysis of three gratings in the crystallized state at three wavelengths (450 nm, 545 nm, and 660 nm) with a thickness error range of 20 nm. The obtained reflection spectra are shown in Figure 13a. For a clearer assessment of the effect of the thickness error, the obtained reflectance spectra were represented on a CIE 1931 chromaticity diagram, as shown in Figure 13b. The results show that the structure of reflecting blue light is most significantly impacted by the thickness error, whereas the structure of reflecting red light displays the least effect.

We also explored the fabrication error sensitivity of the outer diameter *R* of the nanoantenna. The scanning analysis was still performed on the three gratings, and the thickness error range was 5 nm. The obtained reflection spectra are shown in Figure 13c. It can be seen from Figure 13d that the manufacturing sensitivity to the outer diameter *R* of the structure reflecting blue light is the lowest.

## 4. Conclusions

We propose an inverse design using the genetic algorithm to achieve the reversible tuning of Mie resonances in the visible band. The inverse design of the metasurface is achieved through a combination of the genetic algorithm and finite-difference time-domain (FDTD) solutions to generate reflections and color changes at specific wavelengths and achieve reversibly tunable properties by using Sb_2_S_3_. This methodology enables the optimization of parameters and reduces the design cycle, while the nanoscale structure facilitates high integration. Sb_2_S_3_ can be phase transformed via semiconductor laser heating and the electric pulse method of the heating device, where crystallization can be performed using a continuous wave laser and amorphization using a femtosecond laser. Cycling durability can also be improved through a controlled combination of reduced pulse energy and increased pulse number. The results demonstrate that the filter arrays featuring Mie resonance generate significant ∆E of 64.8, 28.1, and 44.1 through switching the phase state of Sb_2_S_3_, and the color change under different incident angles is still much higher than the minimum standard distinguishable by the human eye. This work lays the groundwork for the application of Sb_2_S_3_ in visible light and provides a smart approach for the development of controllable structural colors.

## Figures and Tables

**Figure 1 nanomaterials-13-01452-f001:**
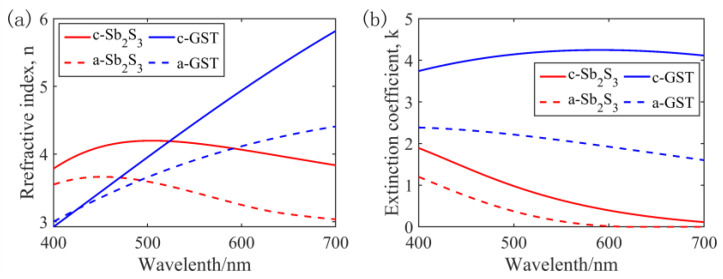
Refractive index *n*, *k* curves of Sb_2_S_3_ (**a**) and GST (**b**).

**Figure 2 nanomaterials-13-01452-f002:**
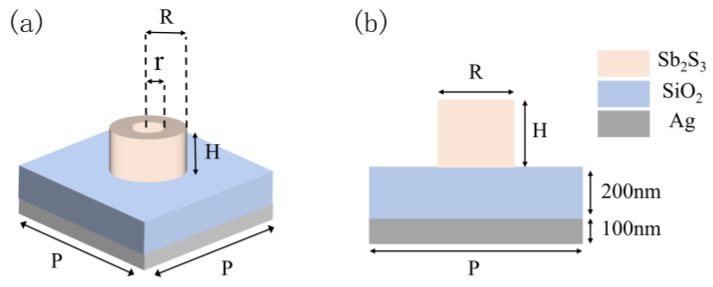
(**a**) Unit structure diagram; (**b**) cross-sectional view of the unit structure.

**Figure 3 nanomaterials-13-01452-f003:**
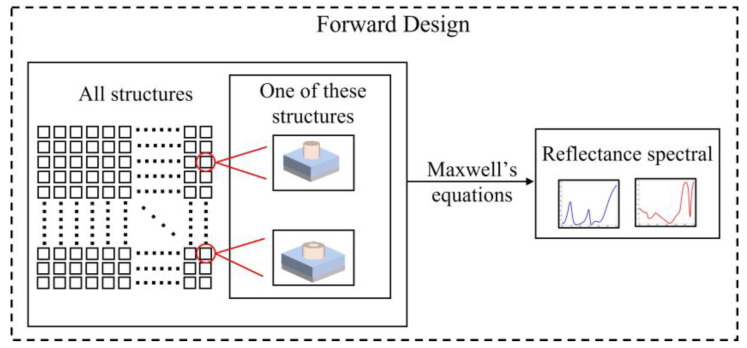
The process of traditional forward design.

**Figure 4 nanomaterials-13-01452-f004:**
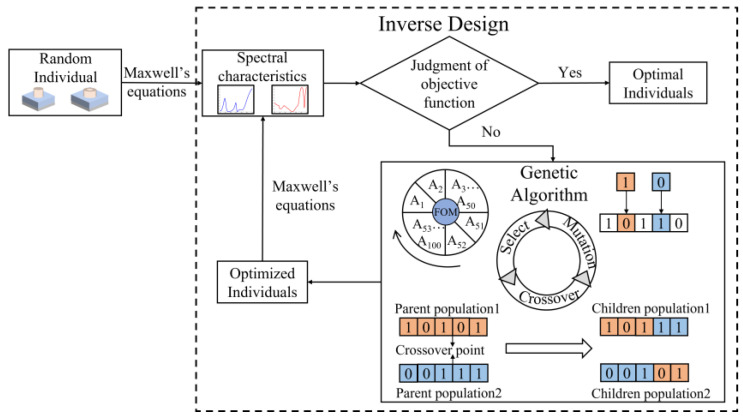
Inverse Optimization Process of Genetic Algorithm for Dielectric Nanostructure.

**Figure 5 nanomaterials-13-01452-f005:**
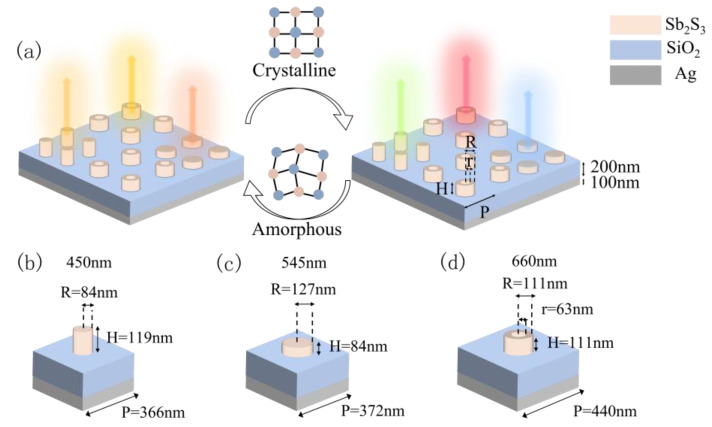
The final design structure diagram of the color filter array. (**a**) Metasurface Periodic Structure Diagram. Detailed three-dimensional structure diagram of the device at three wavelengths of 450 nm (**b**), 545 nm, (**c**) and 660 nm (**d**).

**Figure 6 nanomaterials-13-01452-f006:**
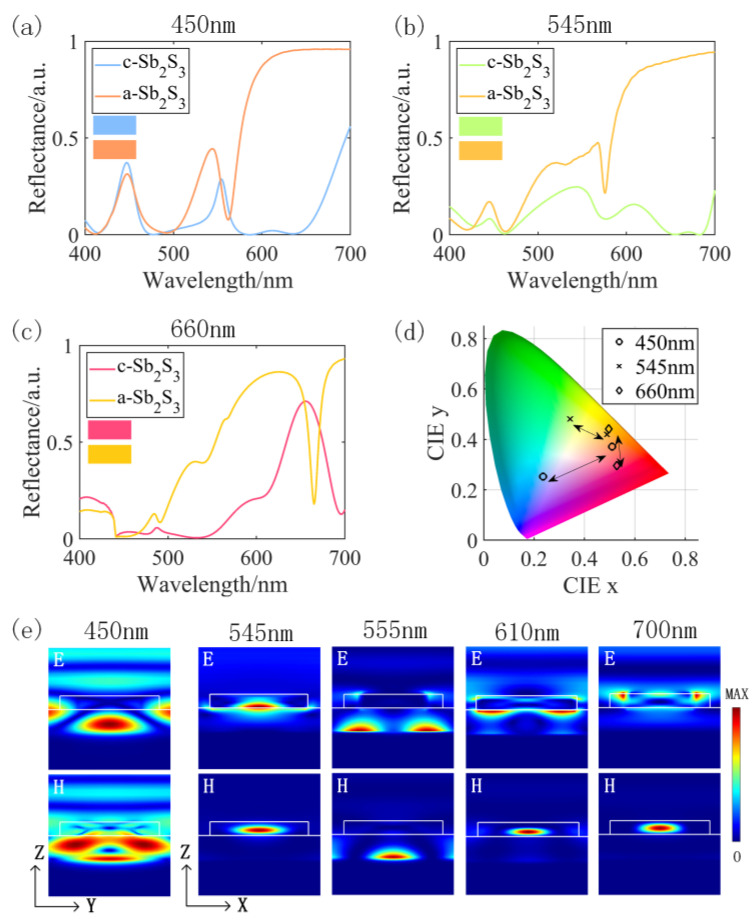
The reflection spectra of the resonance peaks at 450 nm (**a**), 545 nm, (**b**) and 660 nm (**c**) in the crystallized and amorphous states; (**d**) coordinates of reflectance spectra in different states; (**e**) electric and magnetic field strengths on the profile at different peak wavelengths.

**Figure 7 nanomaterials-13-01452-f007:**
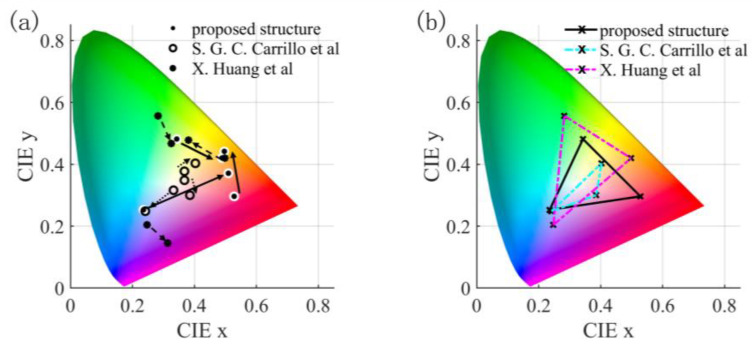
Comparison of color gamut (**a**) and ∆E (**b**) with other work.

**Figure 8 nanomaterials-13-01452-f008:**
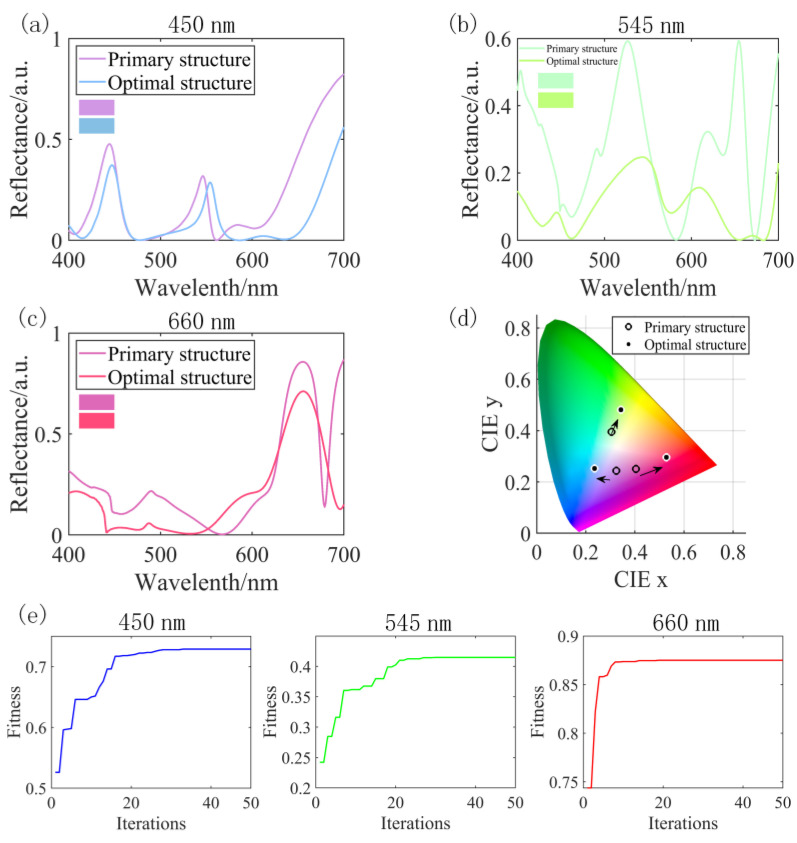
Comparison of reflectance spectra of the structures obtained from the initial population and the optimal population at target wavelengths of 450 nm (**a**), 545 nm, (**b**) and 660 nm (**c**); (**d**) coordinates of reflection spectra under different structures; (**e**) the convergence process of the maximum fitness value during the optimization process for metasurface structures with different resonance peaks.

**Figure 9 nanomaterials-13-01452-f009:**
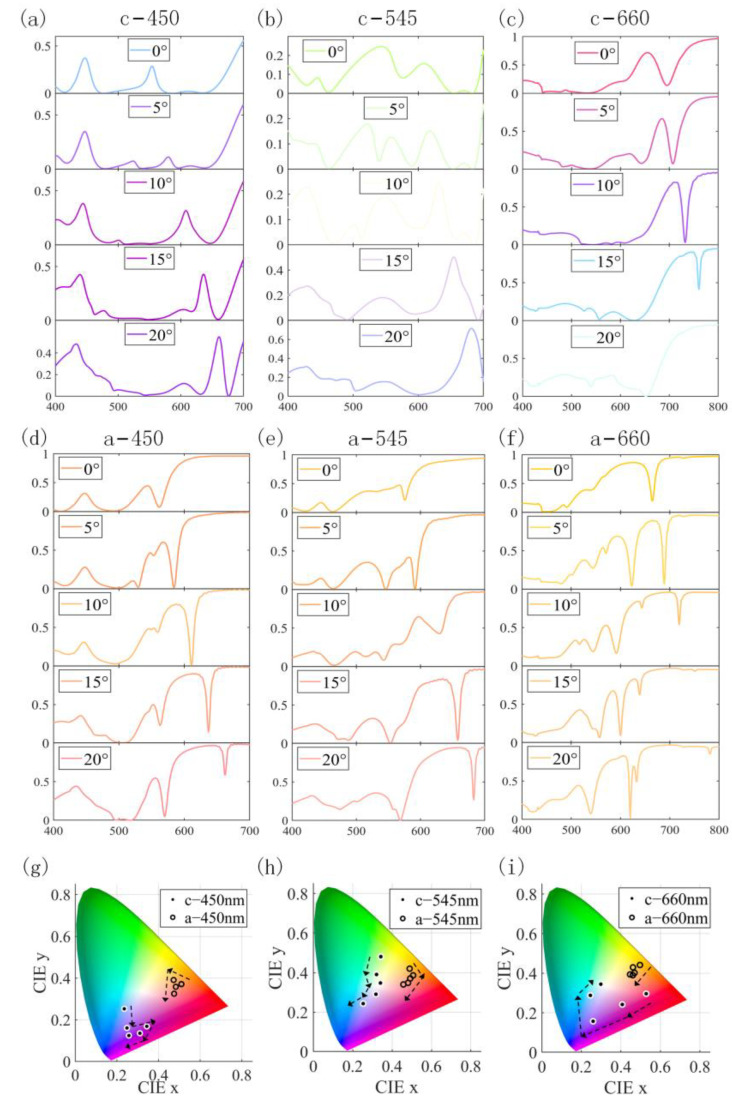
(**a**–**c**): Comparison of reflection spectra at different incident angles in Sb_2_S_3_ crystallized state; (**d**–**f**): comparison of reflection spectra at different incident angles in Sb_2_S_3_ amorphous state; (**g**–**i**): the relationship between incident angle and reflected color.

**Figure 10 nanomaterials-13-01452-f010:**
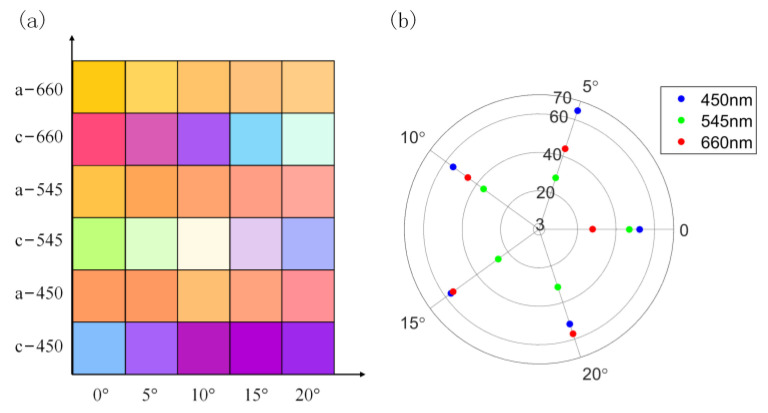
(**a**) Color display of different states under different incident angles; (**b**) the relationship between incident angle and ∆E.

**Figure 11 nanomaterials-13-01452-f011:**
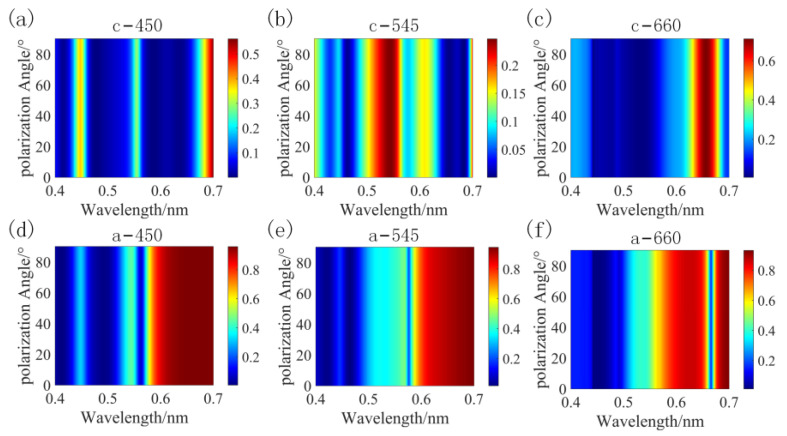
Sensitivity of crystallized structures (**a**–**c**) and amorphous structures (**d**–**f**) to polarization angle.

**Figure 12 nanomaterials-13-01452-f012:**
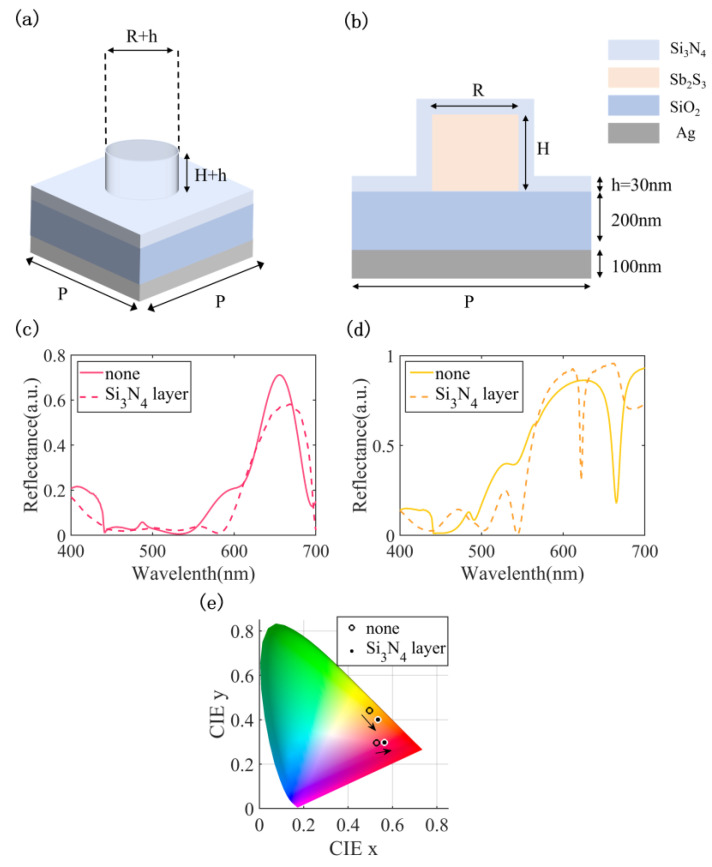
(**a**) The diagram of the unit structure after the addition of the Si_3_N_4_ encapsulation layer; (**b**) the cross-sectional view of the unit posts the incorporation of a Si_3_N_4_ encapsulation layer having a thickness of 30 nm; reflectance curves for structures reflecting red light in the crystallized state (**c**) and amorphous state (**d**) with and without the Si_3_N_4_ encapsulation layer; (**e**) characterization of reflectance curves on the CIE 1931 chromaticity diagram.

**Figure 13 nanomaterials-13-01452-f013:**
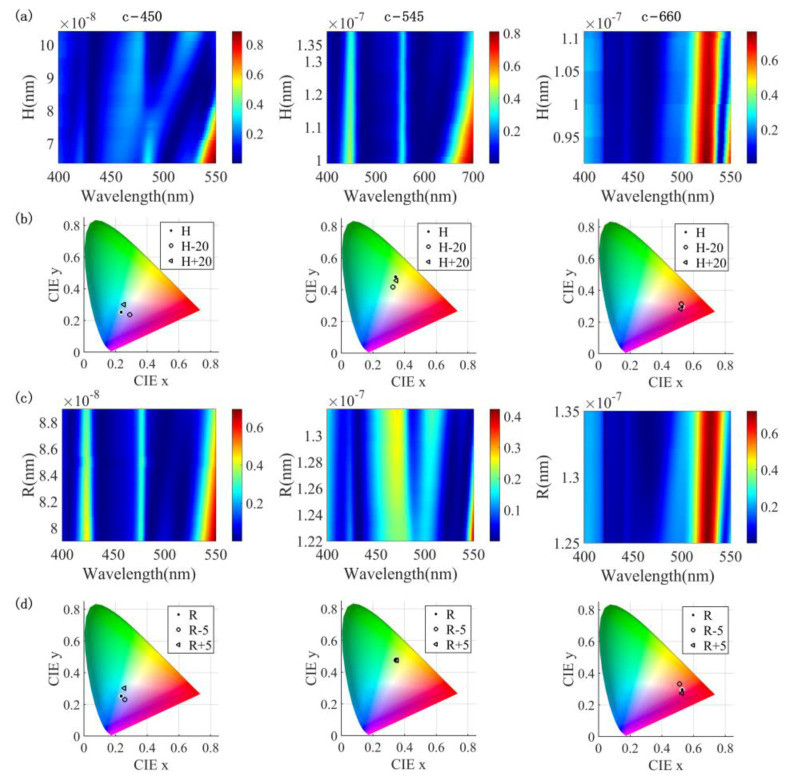
(**a**) Sensitivity to thickness errors in the crystal structure of Sb_2_S_3_ at 450 nm, 545 nm and 660 nm; (**b**) the reflection curves for a thickness error of 20 nm are represented on the CIE 1931 chromaticity diagram; (**c**) sensitivity to outer diameter *R* errors in the crystallized structure of Sb_2_S_3_ at 450 nm, 545 nm, and 660 nm; (**d**) the reflection curves for an outer diameter *R* error of 5 nm are represented on the CIE 1931 chromaticity diagram.

**Table 1 nanomaterials-13-01452-t001:** Optical properties of phase-change materials.

PCM	Conversion Temperature	Melting Point	Bandgap	Switching Time	Phase Change Cyclicity	∆n at 633 nm	k_c_ at 633 nm	∆n/k_c_ at 633 nm
	/K	/K	/eV	/s	/Frequency			
GSST [33]	673	900	—	—	10^5^	0.16	3.82	0.04
GST [34,35]	453	889	0.4~0.7	50 × 10^−9^	10^15^	0.62	2.56	0.24
Sb_2_S_3_ [15,32,36,37,38,39,40,41,42,43,44]	573	823	1.7~2	80 × 10^−9^	>7000	0.87	0.56	1.55

**Table 2 nanomaterials-13-01452-t002:** Optimize the specific parameters of the metasurface structures.

Target Wavelength	Height of Circle	Outer Diameter of Circle	Inner Diameter of Circle	Substrate Period
/nm	*H*/nm	*R*/nm	*r*/nm	*P*/nm
450	119	84	0	366
545	84	127	0	372
660	111	130	63	440

## Data Availability

Data are available in a publicly accessible repository.

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
