# Peer review of "Sb2S3-Based Dynamically Tuned Color Filter Array via Genetic Algorithm"

_nanomaterials, 2023, doi:10.3390/nano13091452_

Round 1
Reviewer 1 Report
X. Wei et al. proposed a method to design Sb2S3 nanostructures for tunable structural coloration using genetic algorithms (GA). This GA-based optimization efficiently screen and specify the geometries of nanostructures that not only display vivid colors, but also maximized color changes under the phase transition. The performance comparison with other works highlights the soundness of their design method and the material selection. Further analysis of incident and polarization angle dependence was conducted. Lastly, they further explore the effects of additional silicon nitride packing and inaccurate induced by fabrication error.
It would have been great if the authors had conducted the following experiments. Regarding the experimental aspect, It would be nicer if the authors fabricate the devices (preparing Sb2S3 film, and pattern it, etc etc) and measure the spectral images using microscopic setup, which surely improve the quality of this paper. Now, this work is purely done with numerical simulation and optimization algorithms.
However, I am afraid I can’t ask for doing so because it certainly delays the publication for so long, also not sure the authors are able to access such fabrication facilities now. They mentioned that the fabrication and further experiments are ongoing as next project, so I’m fine with it.
Nevertheless, I believe the paper is scientifically sound and has novelty, meeting up the stand of Nanomaterials.
Plus, the paper provides interesting approaches and results on optimization of tunable nanostructure so that I suggested the publication of this paper in its current form.
Reviewer 2 Report
This paper describes tunable color filter array with metasurface structure based on phase change material Sb2S3. The authors demonstrated the large chromatic aberration ΔE modulation of 64.8, 28.1, 44.1 by simulation. It would be of interest to Nanomaterials readers, however, I have some major concerns as follows:
1) Figures 8, 9, 12, and 13 are missing.
2) The authors should reconsider the title of this paper. Generic algorithm is no more than a means. “Phase change material Sb2S3” or “large chromatic aberration” shall be included in the title.
3) Page 4, line 133. The meaning of “circular or cylindrical” is unclear. Is it correct that the structure with the inner diameter r=0 is a solid rod, while that with r≠0 is a hollow pipe?
4) It would be informative if the authors mention the sensitivity to fabrication errors. How the reflection spectra and color properties are affected if the size of the structure is slightly varied?
Followings are minor deficiencies:
1) Eq. 13. “3√t, t>l3” should read “3√t, t<l3”.
2) Fig. 7(b). The label of the vertical axis is missing.
1) Table 2. “nm” should not be italic but upright.
2) Fig. 6. In the horizontal axis, “Wavelenth” should read “Wavelength”.
3) Page 12, line 374. “mW” should not be italic but upright.
4) Page 12, line 375. “Li Lu et al” should be “Li Lu et al.” (A period is necessary.)
5) Page 13, line 385. “660nm” should be “660 nm”. (A space is necessary.)
Reviewer 3 Report
• What is the main question addressed by the research?
The main question is answered. However, the novelty of the paper is not good.
• Do you consider the topic original or relevant in the field? Does it address a specific gap in the field?
Topic is 60 percent relevant.
• What does it add to the subject area compared with other published material?
The comparison table need to be revised with more deterministic parameters.
• What specific improvements should the authors consider regarding the methodology? What further controls should be considered?
Methodology is not properly defined. The novelty is missing.
• Are the conclusions consistent with the evidence and arguments presented and do they address the main question posed?
Conclusion need to be revised and it should be precise.
• Are the references appropriate?
Could add more detailed reference regarding meta surfaces.
• Please include any additional comments on the tables and figures.
Figure quality is poor. Table need to be revised.
There are lot of grammatical sentences that need the correction. Better to have professional english correction.
Round 2
Reviewer 3 Report
Thank you for the changes. The manuscript is now good.